# Self-cleavage of the GAIN domain of adhesion G protein-coupled receptors requires multiple domain-extrinsic factors

Yin Kwan Chung [1,2], Christian H. Ihling[3,4], Lina Zielke [1], Signe Mathiasen[2], Andrea Sinz [3,4] & Tobias Langenhan [1,5,6] ✉

The autoproteolysis-inducing (GAIN) domain of class B2/adhesion G protein-coupled receptors (aGPCRs) is structurally conserved, and its self-cleavage is central to receptor mechanotransduction and signaling. Yet, the influence of factors beyond the protein fold on GAIN domain autoproteolysis remains unclear. Using ADGRE2/EMR2, a self-cleaved aGPCR, we investigated contributions of the seven-transmembrane (7TM) region to GAIN domain autoproteolysis during receptor maturation and trafficking. Retention Upon Selective Hook (RUSH) assays showed that self-cleavage acts as a checkpoint before endoplasmic reticulum (ER) exit, but not during plasma membrane transport. Stepwise truncations of the 7TM domain revealed that cleavage can occur before or at synthesis of the first transmembrane helix, and is enhanced with formation of the full 7TM domain. Analyses of six additional cleavage-competent aGPCRs demonstrated that ER membrane tethering facilitates GAIN domain processing, supported by proteomic evidence linking cleavage to proximity with the *N*-glycosylation pathway. These results highlight the interplay between GAIN and 7TM domains, offering mechanistic insights and guiding pharmacological strategies to modulate aGPCR activation and signaling.

Adhesion G protein-coupled receptors (aGPCRs) constitute a large yet poorly understood family within the GPCR superfamily. Owing to their unique architecture of large extracellular regions (ECR) that contain an array of adhesion motifs, and their distinct activation mechanisms, aGPCRs are regarded as a mechanosensor which transduces extracellular mechanical stimuli into intracellular signals[1,2]. This function is required in various physiological processes, such as myotube hypertrophy (ADGRG1), synapse formation (ADGRL2, ADGRL3, ADGRB3), myelination (ADGRG6), secretion of pulmonary surfactants (ADGRF5), functioning of immune cells such as leucocytes and dendritic cells (ADGRE5) and neuronal mechanosensation (ADGRL1-3)[3].

The unifying feature of aGPCRs is a conserved extracellular fold termed the GPCR autoproteolysis-inducing (GAIN) domain[4]. GAIN domain cleavage at a GPCR proteolysis site (GPS) leads to the formation of N- and C-terminal fragment (NTF/CTF) protomers that remain non-covalently associated[5–7]. The newly formed N-terminus of the CTF contains a tethered agonist/*Stachel* sequence, an N-terminal motif of residues that is sufficient and required to stimulate receptor activity by binding to an orthosteric pocket within the 7TM domain of CTF[8–10]. Recent structural and biochemical evidence show an interplay between GAIN domain conformations, GAIN and 7TM domain interactions, and the signalling capability of the receptors[11,12]. Thus,

[1]Rudolf Schönheimer Institute of Biochemistry, Division of General Biochemistry, Medical Faculty, Leipzig University, Leipzig, Germany. [2]Department of Biomedical Sciences, Faculty of Health and Medical Sciences, University of Copenhagen, Copenhagen, Denmark. [3]Department of Pharmaceutical Chemistry and Bioanalytics, Martin-Luther-University Halle-Wittenberg, Halle, Germany. [4]Center for Structural Mass Spectrometry, Martin-Luther-University Halle-Wittenberg, Halle, Germany. [5]Comprehensive Cancer Center Central Germany, Leipzig University, Leipzig, Germany. [6]Institute of Biology, Faculty of Life Sciences, Leipzig University, Leipzig, Germany. ✉e-mail: tobias.langenhan@gmail.com

understanding the mechanism of GAIN domain cleavage is required to fully delineate the signalling pathway of aGPCRs.

The putative mechanism of GAIN domain proteolysis was initially postulated based on a mutagenesis study using the extracellular region (ECR) of the immune receptor ADGRE2/EMR2 (E2) fused with an Fc stalk[13]. Self-cleavage occurs autonomously and automatically through an N → O acyl shift between the GPS residues similar to the autoproteolytic mechanism of Ntn hydrolases[13,14], which was supported by structural evidence[4]. This definition, however, has led to the assumption that the efficiency of the cleavage reaction of the GAIN domain in individual full-length receptors is independent of other parameters. In fact, recent studies have also revealed that GAIN domains display high conformational flexibility[7,15,16], and several observations show that their self-cleavage is context-dependent[4,17–19]. Hence, in addition to structural requirements within the GAIN domain[20,21], its conformation allowing for the auto-proteolytic event is critical and may require domain-extrinsic conditions, e.g., the 7TM domain of the receptor molecule that is localised in close proximity. Although the structures of GAIN domains and 7TMs have been individually obtained[22,23], the full-length structure of only one receptor, ADGRE5/CD97, has been elucidated[11]. In this structure, the GAIN domain is oriented almost perpendicularly to the 7TM domain, establishing direct contacts with the first extracellular loop (ECL1) and ECL2[11], which contrasts with another study in ADGRL3 GAIN-7TM structural reconstruction where the GAIN domain remains parallel to the membrane surface[24].

The influence of GAIN domain cleavage on the subcellular trafficking and localisation of aGPCRs is also controversially discussed. It was observed for polycystin-1 (PC1)-like proteins, the only other molecule family that possesses the GAIN fold, that GAIN domain cleavage is obligatory for endoplasmic reticulum (ER) exit and trafficking to the designated location[25]. In contrast, other results have provided evidence that cleavage of PC1 and auto-proteolysable aGPCRs is not necessary for their entry into the trafficking route[13,26–28].

Here, we examined the consequences of GAIN domain cleavage on the plasma membrane delivery of aGPCRs, and explored determinants that contribute to it throughout receptor synthesis and maturation. Using E2 as a model cleavage-competent aGPCR, we show that E2 self-cleavage occurs before exit from the ER and is not required for subsequent trafficking within the secretory pathway. We find that the 7TM domain and membrane proximity aids GAIN domain cleavage through tethering the GAIN domain to the lumenal side of the ER membrane during the biogenesis. GAIN domain-specific photo-cross-linking followed by mass spectrometric (MS) analyses of its immediate vicinity reveals its molecular neighbourhood, including enzymes of the N-glycosylation pathway, which act as autoproteolysis facilitators.

## Results

### TM1 of E2 is essential for efficient GAIN domain cleavage via tethering to the lumenal side of the ER membrane

Cumulative evidence has suggested that the GAIN domain cleavage occurs before ER exit for some aGPCRs, including E2 (Fig. 1a)[5,13,28]. However, it is unknown when GAIN domain cleavage occurs during or after receptor translation in the ER. Thus, folding of the GAIN domain and subsequent self-cleavage can occur before, during or after completion of 7TM domain biogenesis (Fig. 1b).

To clarify the relative timepoint of GAIN domain cleavage, we constructed a series of E2 7TM domain truncations wherein increasing numbers of TMs are present in N → C order (E2^WT-xTM), to stall the biosynthesis of the 7TM domain after each TM helix and determine the state of GAIN domain autoproteolysis (Fig. 1b). All E2^WT-xTM mutants were well expressed in HEK293T cells at similar levels as determined by ELISA (Fig. 1c; all receptor versions tagged with an N-terminal HA-tag) and Western blotting, which also showed that E2^WT-7TM and the TM truncation mutants were heavily glycosylated (Fig. 1d). Previous

observation suggested that E2 is only N-glycosylated but not O-glycosylated[13]. Thus, we treated lysates with PNGase F, a non-specific N-glycosidase, to reduce the glycosylated protein signals into a single band (Fig. 1e; Supplementary Table 1). A-50 kDa band after PNGase F treatment was observed in the cleavage-competent E2^WT-7TM protein (Fig. 1f). A cleavage-resistant E2^H516A-7TM control protein, in which the E2 GPS at the −2 (His^516) position was replaced by alanine, confirmed the mass of the FL receptor and showed no NTF (Fig. 1f). Interestingly, NTF generation was observed in all PNGase F-treated truncation mutants (Fig. 1f), indicating that NTF generation through GAIN domain cleavage occurs before or during TM1 translation. These observations lead to several follow-up conjectures (Fig. 1g). First, the GAIN domain is self-sufficient for proteolysis as cleavage occurs independently of the translation of the C-terminal parts of the receptor protein, including the 7TM domain, as long as the GAIN domain is folded correctly. Second, if this is not the case, efficient GAIN domain self-cleavage requires ER membrane tethering provided by specific sequences in the E2 TM1 helix or, third, the helical fold of TM1 itself.

We generated chimeric E2 receptor protein fusions that are retained in the ER (confirmed by confocal microscopy in Supplementary Fig. 1a, b), one with (E2-ECR-CalnTM^ER, ECR denotes the extracellular region) and one without an ER membrane anchor (E2-ECR-tdT-KDEL^ER), and compared their cleavage efficiencies to each other (Fig. 1h). As GAIN domain cleavage is a binary event (an individual receptor protein can either be cleaved or not), the uncleaved(FL):cleaved(NTF) protein ratio of expressed receptor molecules captures the efficiency of GAIN domain autoproteolysis. E2-ECR-CalnTM^ER consisted of the E2-ECR and the single TM helix of calnexin, an ER-resident protein that attaches the fusion protein to the lumenal side of the ER membrane, preventing its exit (Fig. 1h, Supplementary Fig. 1a)[29]. The soluble E2-ECR-tdT-KDEL^ER protein was generated by fusion of the E2-ECR to a 2xtdTomato cassette, and targeted to the ER lumen by a C-terminal KDEL sequence for ER retention (Fig. 1h, Supplementary Fig. 1b)[30]. E2-NTF held in the ER (E2-NTF-KDEL^ER) served as protein size control (Fig. 1h). Each protein was N-terminally HA-tagged. If the GAIN domain is self-sufficient for the cleavage reaction, E2-ECR-tdT-KDEL^ER, devoid of membrane anchorage, would show an extent of proteolysis similar to E2-ECR-CalnTM^ER.

Upon expression in HEK293T cells, GAIN domain cleavage of E2^WT-ECR-CalnTM^ER produced an NTF with similar mobility to E2-NTF-KDEL^ER (Fig. 1i, k). Cleavage-resistant E2^H516A-ECR-CalnTM^ER corroborated the identity of the E2^WT-ECR-CalnTM^ER NTF band (Fig. 1i). In contrast, GAIN domain cleavage of the non-tethered E2^WT-ECR-tdT-KDEL^ER was strongly reduced (Fig. 1i, k). Western blotting of E2^WT-ECR-tdT-KDEL^ER showed two intense protein bands at -125 kDa (representing different glycosylation states), similarly to cleavage-resistant E2^H516A-ECR-tdT-KDEL^ER (Fig. 1i), and were recognised by their N- and C-terminal tags, further corroborating the lack of GAIN domain cleavage (Fig. 1j). Only a small amount of NTF from E2^WT-ECR-tdT-KDEL^ER was observed (Fig. 1i-j).

As E2^WT-ECR-CalnTM^ER is considerably self-cleaved (Fig. 1k), it shows that no specific sequences in the E2 TM1 are necessary for the proteolysis. We also replaced the CalnTM with the TM region of platelet-derived growth factor receptor (E2-ECR-PDGFR-TM), which allows for efficient targeting of the chimeric E2-ECR-PDGFR-TM protein to the plasma membrane (Supplementary Fig. 1c). E2^WT-ECR-PDGFR-TM also showed efficient self-cleavage, which was abolished in a cleavage-deficient variant (Supplementary Fig. 1d, e). Thus, E2 GAIN domain cleavage is strongly favoured by ER membrane tethering through TM1 during receptor biogenesis. (Fig. 1l).

### The E2 7TM domain facilitates GAIN domain cleavage

Although two-thirds of aGPCR homologues are found or predicted to be GAIN domain cleavable, different expression conditions affect cleavage efficiency[4,17–19]. Quantification of FL:NTF ratios based on

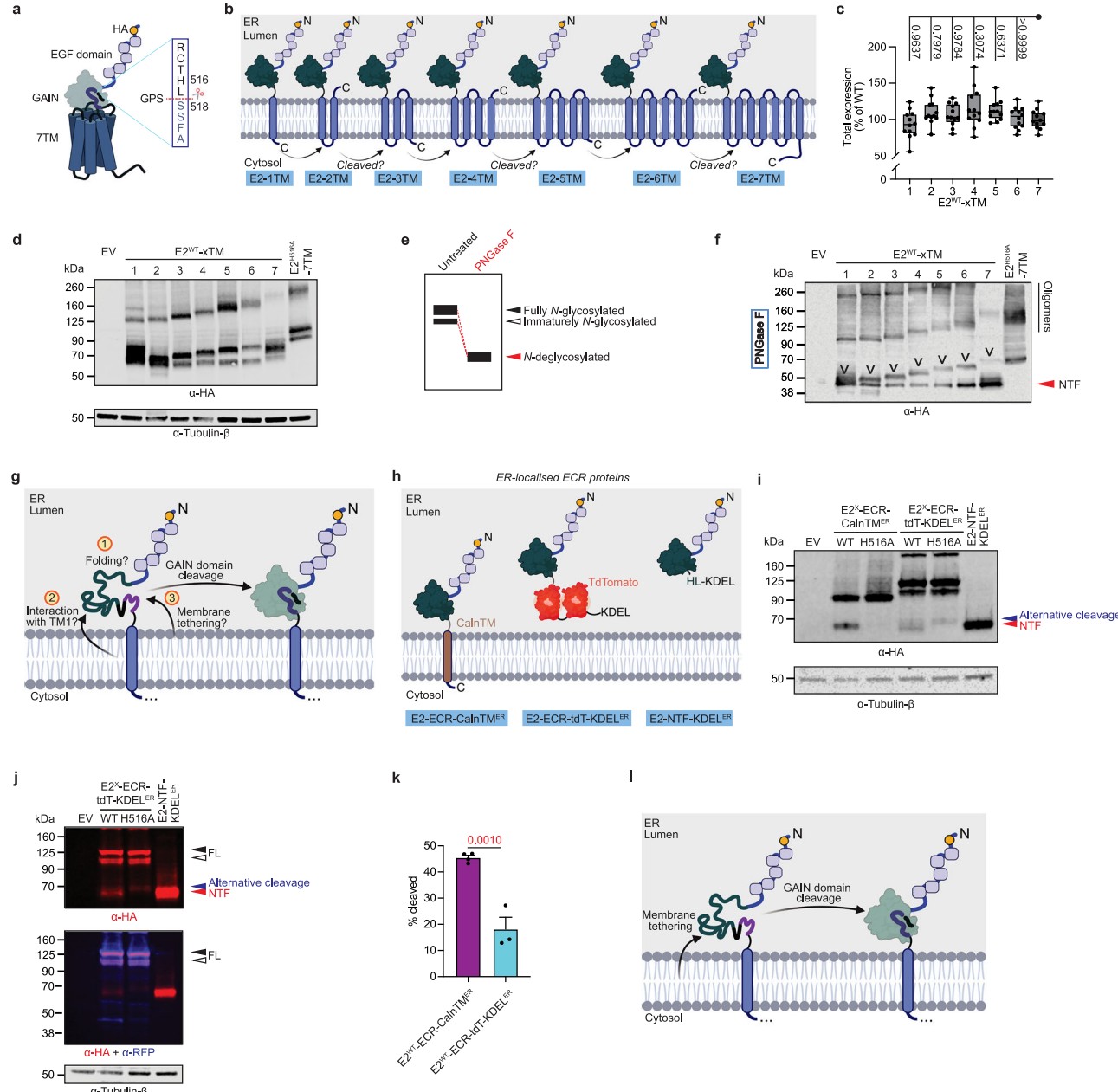

Western blots suggested that E2$^{WT}$-7TM was predominantly (~90 %) cleaved when compared with proteolysis-deficient E2$^{H516A}$-7TM (Figs. 1f and 2a). Albeit the E2 TM truncation mutants were capable of autoproteolysis, only approximately half of the E2$^{WT}$-3TM and E2$^{WT}$-4TM proteins self-cleaved (Fig. 2a; blots of E2$^{WT}$-1TM and E2$^{WT}$-2TM were not quantifiable due to the failure of resolution between the FL and NTF), while ≥ 4 TM helices resulted in cleavage comparable to E2$^{WT}$-7TM (Fig. 2a). This suggests that the 7TM domain stabilises a cleavage-competent GAIN domain conformation during 7TM biogenesis[7,31].

If E2 autoproteolysis occurs via specific intermolecular interactions between the GAIN and 7TM domains, notably the ECLs (Fig. 2b), the extent of cleavage would be sensitive to changes in the loops' sequences. To test this, we fused the ECR of E2 with the 7TM regions of different receptors (Fig. 2c). These included several receptors from other self-cleavable (L3, D1, G1) (Fig. 2d and Supplementary Fig. 2a) or non-cleavable (B3) aGPCR subfamilies (Fig. 2e and Supplementary Fig. 2b), and class A GPCRs (β$_2$-adrenergic receptor, dopamine D$_2$ receptor, purinergic P2Y$_{12}$ receptor) (Fig. 2f and Supplementary Fig. 2b). Surprisingly, none of the chimeras altered the extent of GAIN

domain autoproteolysis of E2 (Fig. 2d–f, Supplementary Fig. 2a, b). This indicates that the structural scaffold of the 7TM region, rather than a specific sequence encoded within, promotes GAIN domain self-cleavage (Fig. 2g).

## Uncleaved cleavage-competent E2 is retained in the ER

Next, we determined the intracellular location of cleavable E2$^{WT}$-xTM proteins in a quantitative manner. First, we utilised the differential sensitivity of proteins at different stage of the secretory pathway towards glycosidase treatments as guideposts along the ER-Golgi-plasma membrane route[28]. Cell lysates containing different E2$^{WT}$-xTM variants were treated with endoglycosidase H (Endo H), which enzymatically removes N-glycans added to the proteins only before they arrive at the medial Golgi complex (Fig. 3a, b)[32]. This differentiates two subpopulations of E2$^{WT}$-xTMs in Western blotting (Fig. 3a). Endo H-sensitive E2$^{WT}$-xTM proteins (with higher mobility shift) are thus located in the ER or early Golgi network, while Endo H-resistant E2$^{WT}$-xTM (with lower mobility shift) have entered the late Golgi, from which they are passed on to the plasma membrane or the proteasome system

**Fig. 1 | GAIN domain cleavage of E2 requires tethering of the domain to the luminal side of the ER during receptor biogenesis. a** Structure of 7TM variant of E2 (E2-7TM). The adhesion motifs, the GAIN domain and the 7TM region are labelled. GPS sequence and the numbering of the −2 (H516) and +1 (S518) positions, are also shown. Cleavage site is indicated in red dotted line. Created in BioRender. Chung, Y. (2025) https://BioRender.com/iiztpfd. **b** GAIN domain cleavage can potentially occur at any time shortly before, during or after the translation of the 7TM domain as the GAIN domain is positioned N-terminal to the 7TM fold. Created in BioRender. Chung, Y. (2025) https://BioRender.com/e2it7qs. **c** Total expressions of the TM truncation mutants of E2 (E2$^{WT}$-xTM) were verified by ELISA. All data points were plotted and represented as box-and-whisker diagrams (Horizontal lines of the box from bottom to top: 25$^{th}$ percentile, median, 75$^{th}$ percentile; Whiskers: minimum and maximum values). *P*-values are also shown. **d** Expression patterns of E2$^{WT}$-xTM without glycosidase treatment were examined by Western blotting, detected against the N-terminal HA tag. Detection of tubulin-β is also shown as a loading control. **e** The smeary bands in E2 variants because of differential *N*-glycosylation can be resolved by treatment of the lysates with PNGase F (red). PNGase F, as a non-specific *N*-glycosidase, removes all *N*-glycans from the protein molecules, resulting in a single band with higher mobility, indicating a decrease in molecular weight (red dashed lines). **f** E2$^{WT}$-xTM treated with PNGase F were examined by Western blotting, detected against the N-terminal HA tag. Bands indicative of uncleaved monomeric subpopulations are pointed as an inverted caret (ᵥ). Quantification of the extent of GAIN domain cleavage of monomeric E2$^{WT}$-xTM is shown in Fig. 2a. Detection of tubulin-β is unnecessary because the intensity of the cleaved and uncleaved subpopulations should vary proportionally, would there be errors in the loading of the samples. **g** Three possible explanations for the GAIN domain cleavability observed in E2$^{WT}$-1TM. GAIN domain cleavage can (1) occur naturally after proper folding of the GAIN domain in the middle of the first TM generation, or (2) be catalysed by some molecular factor(s) in the first TM of E2, or alternatively (3) the tethering of the folded GAIN domain onto the lumenal side of the ER membrane promotes GAIN domain cleavage. Created in BioRender.

Chung, Y. (2025) https://BioRender.com/rtayw9j. **h** Designs of ER-localised proteins of E2-ECR. The ECR of E2 is either conjugated to an ER-resident calnexin TM (E2-ECR-CalnTM$^{ER}$) or is fused to a double TdTomato moiety localised in the lumen of ER via the KDEL sequence (E2-ECR-tdT-KDEL$^{ER}$). If the two proteins are cleaved at the GAIN domain, an NTF fragment will be produced, showing a band in Western blotting with similar mobility to that of an ER-localised NTF (E2-NTF-KDEL$^{ER}$). Subcellular localisations of E2-ECR-CalnTM$^{ER}$ and E2-ECR-tdT-KDEL$^{ER}$ are shown in Supplementary Fig. 1a–b. Created in BioRender. Chung, Y. (2025) https://BioRender.com/t47b6q3. **i** Expression patterns of the ER-localised E2-ECR proteins were verified by Western blotting, detected against the N-terminal HA tag. Bands representing NTF and alternative cleavage are labelled in red and deep blue respectively. Detection of tubulin-β is also shown as a loading control. Quantification on the extent of cleavage is shown in (**k**). **j** The expression patterns of E2-ECR-tdT-KDEL$^{ER}$ proteins were examined by Western blotting, detected against the N-terminal HA tag (in red) and RFP signal (in deep blue) that also targets TdTomato. The overlap of the two signals (in purple) indicates the uncleaved product. Bands representing NTF, alternative cleavage, and FL are labelled in red, blue and black, respectively. Full and immature glycosylations are indicated by solid and open triangles, respectively. Detection of tubulin-β is also shown as a loading control. Of note: a faint band below the NTF is observed in E2$^{H516A}$-ECR-CalnTM$^{ER}$ and E2$^{H516A}$-ECR-tdT-KDEL$^{ER}$ suggesting additional proteolytic processing of E2 in proximity to GPS (Krasnoperov et al.[56]). **k** Quantification on the extents of cleavage of ER-localised E2-ECR proteins. The extent of cleavage is calculated as the fraction of the intensity of the NTF over the total intensity of NTF and FL. Data are shown as mean ± SEM. All individual values are plotted. *P*-values are also shown and highlighted in red when it is below the significance level (95%). **l** TM1 of E2 assists with GAIN domain cleavage via tethering of the domain to the luminal side of the ER membrane. Created in BioRender. Chung, Y. (2025) https://BioRender.com/c03ujh9. Source data are provided as a Source Data file. See Methods section for statistics and reproducibility.

---

(Fig. 3a, b). As E2 is only *N*-glycosylated, Endo H-sensitive bands would migrate to the same position as when they are treated with PNGase F, while the mobility of Endo H-resistant bands would be lagged (Fig. 3a). This facilitates our annotations on the Endo H-sensitivities of the E2$^{WT}$-xTM proteins (Fig. 3b and Supplementary Fig. 3a).

Expression of E2$^{WT}$-7TM produced two NTF bands of ~70 kDa (Endo H-resistant) and ~50 kDa (Endo H-sensitive) upon enzyme treatment (Fig. 3b), with the majority of the receptor protein being Endo H-resistant (Fig. 3b, c). All truncation mutants were Endo H sensitive, but some displayed also Endo H-resistant bands indicating a selective capability to enter the medial Golgi complex (Fig. 3b, c and Supplementary Fig. 3a). The highest Endo H resistance was observed for E2$^{WT}$-3TM and E2$^{WT}$-5TM, although at much lower level than the E2$^{WT}$-7TM control (Fig. 3b, c and Supplementary Fig. 3a). In contrast, E2$^{WT}$-4TM and E2$^{WT}$-6TM were almost non-resistant to Endo H processing (Fig. 3c). The surface expression of the TM truncation mutants followed a similar trend consistent with the relative degrees of Endo H resistance observed in the receptor monomers (Fig. 3d). Importantly, the uncleaved subpopulation of all the E2$^{WT}$-xTM proteins (in both monomeric and oligomeric states) were not Endo H-resistant and, thus, retained in the ER or cis-Golgi (Fig. 3b and Supplementary Fig. 3a).

This was confirmed by immunocytochemical labelling and confocal imaging of the receptor variants (Supplementary Fig. 3b-d). E2$^{WT}$-7TM was largely found at the plasma membrane (Supplementary Fig. 3d), E2$^{WT}$-1TM, which also produced a prominent Endo H-resistant NTF band (Fig. 3b), was observed in both the ER and the plasma membrane, but not the Golgi (Supplementary Fig. 3b). E2$^{WT}$-4TM, which displayed low GAIN cleavage efficiency and low Endo H-resistance (Fig. 2a–c), was almost entirely retained in the ER and absent from the cell surface (Supplementary Fig. 3c). We conclude that the uncleaved subpopulation of the cleavage-competent E2$^{WT}$-4TM receptor, and by extension also other uncleaved E2$^{WT}$-xTM proteins, are unable to exit the ER.

## GAIN domain cleavage facilitates E2 receptor entry into the secretory pathway

These findings suggest that GAIN domain cleavage affects trafficking of the receptor from the ER to the Golgi. To follow this up, we overexpressed cleavage-competent (E2$^{WT}$-7TM) and -deficient (E2$^{H516A}$-7TM, E2$^{S518A}$-7TM) E2 proteins (Fig. 1a) and found no difference in surface and total expression (Fig. 3e, f). α-HA Western blots confirmed that GPS mutations completely suppressed GAIN domain cleavage (Fig. 3g). Such steady-level expression, however, precludes analyses on how autoproteolysis impacts different timepoints during protein synthesis, delivery and turnover. Thus, the exact assessment of how GAIN domain cleavage affects these processes is very limited[33].

To improve the spatiotemporal resolution of protein trafficking studies, we adopted the Retention Upon Selective Hook (RUSH) assay, pioneered for class A GPCR trafficking studies[34], to follow E2 in the secretory pathway. A KDEL-mediated ER-localised streptavidin component (hook) constitutively retains a cargo receptor in the ER lumen through an N-terminal fusion to a streptavidin-binding peptide (SBP) (Fig. 3h). Addition of the high-affinity streptavidin binding partner biotin outcompetes the SBP-hook interaction and triggers synchronised receptor release from the ER to the surface trafficking pathway (Fig. 3h).

First, we generated a SBP-eGFP-E2$^{WT}$-7TM fusion protein and followed its trafficking using confocal microscopy (Fig. 3i, j, Supplementary Movie 1, 2). Without biotin, SBP-eGFP-E2$^{WT}$-7TM was primarily localised in the ER. Upon biotin addition, SBP-eGFP-E2$^{WT}$-7TM rapidly trafficked to the Golgi apparatus, and plasma membrane residence was observed after approximately 1 hour of biotin treatment (Fig. 3i, j). This demonstrates that the RUSH assay can be used to temporally control of E2 entry in the secretory pathway. Prolonged residence of the receptor in the ER compartment, as a result of the hook-mediated retention, did not alter the extent of cleavage of SBP-E2$^{WT}$-7TM (Supplementary Fig. 4a, b).

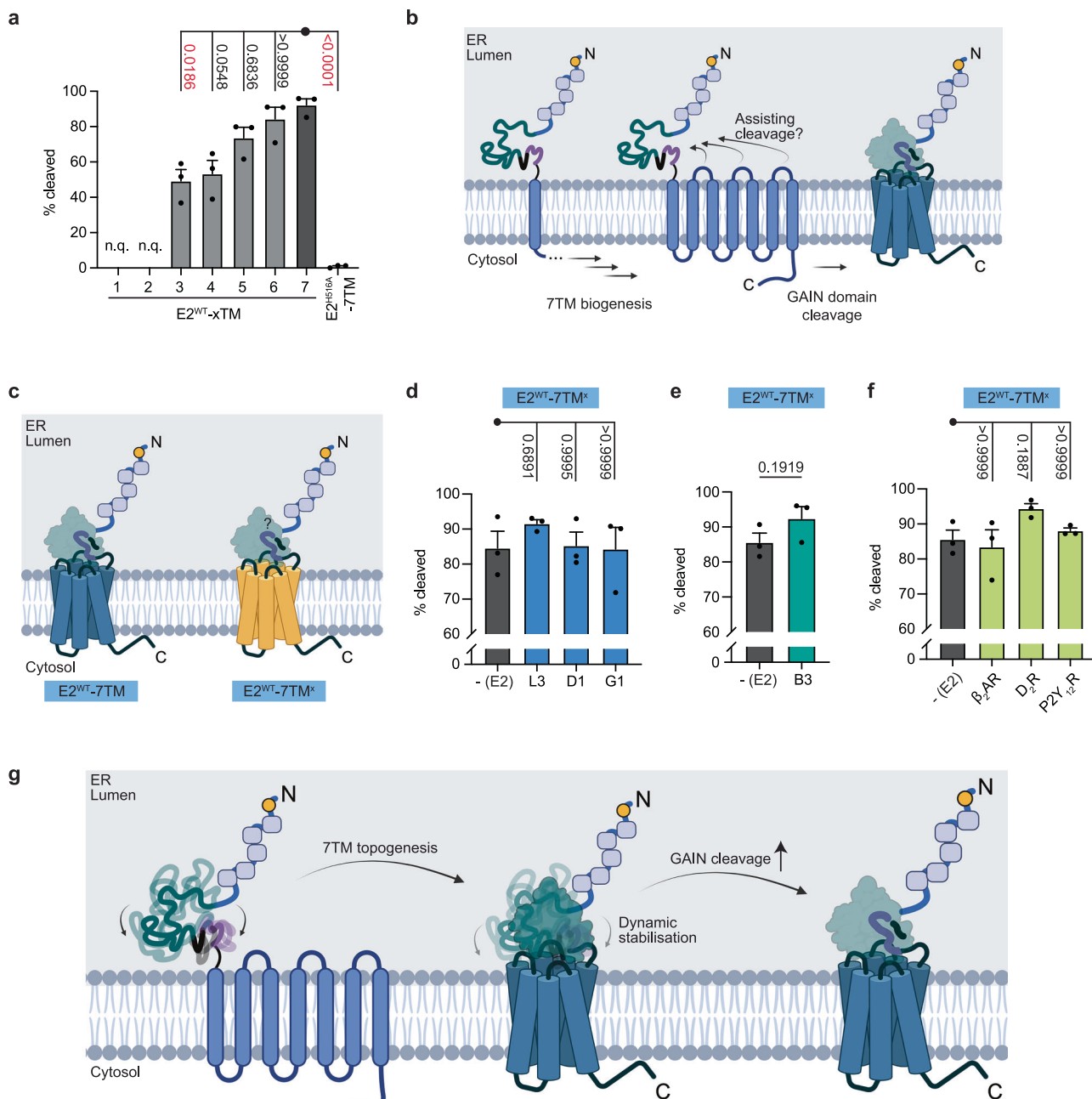

**Fig. 2 | The 7TM region physically facilitates GAIN domain cleavage of E2.**
**a** Quantification on the extents of cleavage of E2$^{WT}$-xTM, with the representative Western blot shown in Fig. 1f. The extent of cleavage is calculated as the fraction of the intensity of the NTF over the total intensity of NTF and FL of the monomeric receptor population. E2$^{WT}$-1TM and E2$^{WT}$-2TM were not quantifiable (n.q.) owing to the failure in resolutions between the NTF and the FL. Data are shown as mean ± SEM. All individual values are plotted. *P*-values are also shown, and highlighted in red when it is below the significance level (95%). **b** During the 7TM biogenesis, the extracellular cavity may assist in GAIN domain cleavage via molecular interactions between the two moieties. Created in BioRender. Chung, Y. (2025) https://BioRender.com/skzm2to. **c** Exchange of the 7TM domain of E2 with that from other

receptors would determine if the assistance of GAIN domain cleavage from the 7TM region requires the molecular determinants on the innate 7TM. Created in BioRender. Chung, Y. (2025) https://BioRender.com/721xw4b. Quantifications of the extent of GAIN domain cleavage of E2 when the ECR was conjugated to the 7TM regions of **d** several cleavable aGPCRs, **e** a non-cleavable aGPCR, or **f** several Class A GPCRs. Data are shown as mean ± SEM. Individual values are plotted. The *p*-values for each comparison are listed. *N* = 3. Representative Western blots are shown in Supplementary Fig. 2. **g** The topology of the 7TM region stabilises the GAIN domain, which favours GAIN domain cleavage. Created in BioRender. Chung, Y. (2025) https://BioRender.com/6lt9t7r. Source data are provided as a Source Data file. See Methods section for statistics and reproducibility.

Next, we co-expressed hook and SBP-E2-7TM with or without GPS mutations to assess total amount and kinetics of surface delivery for each. All SBP-E2-7TM receptor variants were expressed at similar levels without the addition of biotin (Fig. 3k). We then monitored surface expression levels in intervals for up to 6 hours after biotin treatment. SBP-E2$^{WT}$-7TM and SBP-E2$^{H516A}$-7TM showed a basal ER leakage to the

plasma membrane even without biotin treatment (Fig. 3l). Nonetheless, we observed a prominent increase in surface expression of SBP-E2$^{WT}$-7TM after biotin treatment, indicating that the RUSH assay can be used to chase surface delivery of SBP-E2-7TM versions after the biotin pulse (Fig. 3m). In contrast, all SBP-E2-7TM GPS mutants displayed a lower maximal surface level than SBP-E2$^{WT}$-7TM starting at

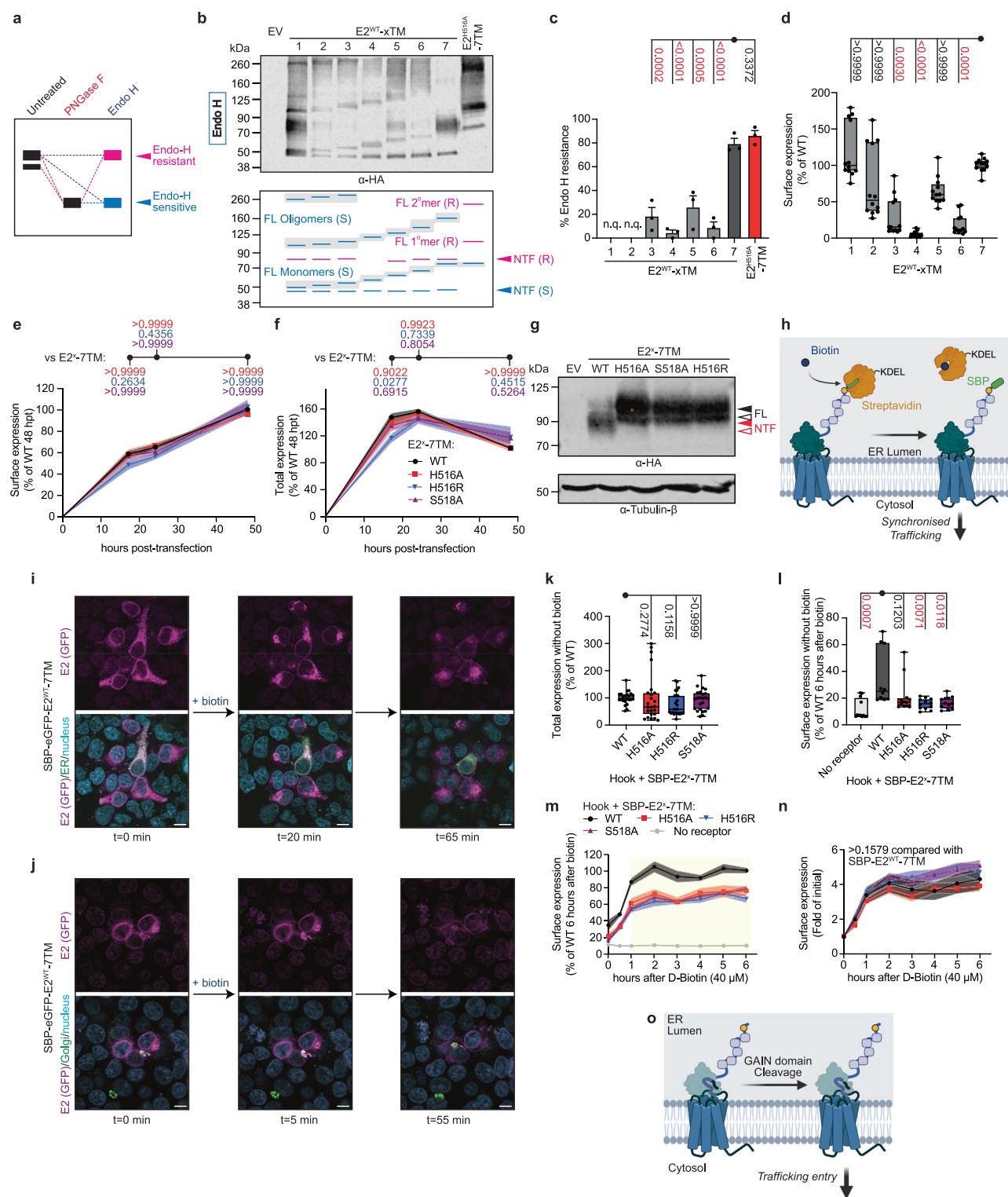

1 hour post biotin treatment (Fig. 3m). However, all autoproteolysis-deficient E2 mutants reached the cell surface at a similar rate to SBP-E2$^{WT}$-7TM, indicating that speed of trafficking is unaffected once the mutants are released from the ER (Fig. 3n). Thus, GAIN domain cleavage is contributory towards controlling the quantity of surface receptor pool, but not the rate of receptor delivery. This suggests that GAIN domain cleavage has exerted its effect before aGPCRs has entered the trafficking route (Fig. 3o).

We finally tested the ability of the arrested E2 TM truncation mutants to enter the trafficking route and measured the synchronised rate of surface delivery of the mutants through the RUSH assay via surface ELISA (Supplementary Fig. 4c). Within the chase period after biotin treatment only SBP-E2$^{WT}$-7TM and SBP-E2$^{WT}$-1TM were able to traffic to the cell surface (Supplementary Fig. 4c) (half-time: 29.4 ± 5.2 min for SBP-E2$^{WT}$-7TM, 19.6 ± 3.4 min for SBP-E2$^{WT}$-1TM; mean ± SEM), while all other truncations did not (Supplementary Fig. 4c), despite a considerable level of expression and extent of cleavage (Supplementary Fig. 4d-f). This is in line with the reduced Endo-H resistance of E2$^{WT}$-(2-6)TM proteins, while E2$^{WT}$-1TM is highly Endo-H resistant (Fig. 3b).

**Fig. 3 | GAIN domain cleavage promotes ER exit of E2. a** The fuzzy bands in E2 variants because of differential *N*-glycosylation can be resolved by treatment of the lysates with Endo H (blue). Endo H only removes *N*-glycans that are attached to the protein before the medial Golgi apparatus. This leads to two situations with Endo H-sensitive/resistant subpopulations (blue dashed lines), providing information on the subcellular localisations of the proteins. As E2 is only *N*-glycosylated, the Endo H-sensitive bands (in blue) will have a similar mobility with the PNGase F-treated bands, while the Endo H-resistant bands (in magenta) have a slower migration. **b**, **c** E2$^{WT}$-xTM treated with Endo H were examined by Western blotting, detected against the N-terminal HA tag. A schematic of the Western blot is provided here, with Endo H-resistant (marked with 'R') and sensitive (marked with 'S') bands coded in magenta and blue, respectively. The uncleaved receptor molecules are shaded in gray. A side-by-side comparison between the blots from PNGase F- and Endo H-treated lysates, which allows the efficient identification of the Endo H-sensitivities of the proteins, is shown in Supplementary Fig. 3a. Quantification of the extent of Endo H resistance of monomeric E2$^{WT}$-xTM, indicative of their entry into the trans-Golgi network, is shown in (**c**). The extent of Endo H resistance is calculated as the fraction of the intensity of the Endo-H resistant band over the total intensity indicating a monomeric receptor. The *p*-values for each comparison are listed. $N = 3$. Subcellular localisations of selected E2$^{WT}$-xTM are shown in Supplementary Fig. 3b-d. Detection of tubulin-β is unnecessary because the intensity of the cleaved and uncleaved subpopulations should vary proportionally, would there be errors in the loading of the samples. **d** Surface expressions of E2$^{WT}$-xTM were verified by surface ELISA. All data points were plotted and represented as box-and-whisker diagrams with parameters as stated in Fig. 1c. The *p*-values for each comparison are listed. HEK293T transiently expressing E2$^{WT}$-7TM and GPS mutants for the indicated durations were subjected to (**e**) surface ELISA and (**f**) total ELISA. Data are expressed as the fraction of the signals from WT after 48 hours of transfection. Data are shown as mean ± SEM indicated in shaded area. The *p*-values for each comparison are listed following the colour codes of the mutants. **g** The expression of E2$^{WT}$-7TM and GPS mutants were examined by Western blotting, detected against the N-terminal HA tag. Bands representing NTF and FL are labelled in red and black, respectively. Full and immature glycosylations are indicated by solid and open triangles, respectively. Detection of tubulin-β is also shown as a loading control. **h** Schematic of the RUSH assay. The biosynthesised receptor is held in ER by the interactions between the SBP (in green) and the ER-localised streptavidin (in orange). The addition of biotin (in blue) triggers the dissociation between the two entities, leading to a synchronised trafficking of the receptor molecules. Created in BioRender. Chung, Y. (2025) https://BioRender.com/0tossi5. Image traces of SBP-eGFP-E2$^{WT}$-7TM trafficking upon biotin addition (40 µM) by confocal microscopy, in which **i** the ER, or **j** the Golgi was also visualised. Time-lapse monitorings of the receptor transport are shown in Supplementary Movies 1-2. The eGFP moiety is inserted after the SBP sequence and before the HA tag in the N-terminus of E2$^{WT}$-7TM, replacing the position for the start Methionine and the innate signal peptide of the receptor. Location of receptor was monitored by the eGFP signal, indicated in purple. ER or Golgi was visualised by CellLight BacMam 2.0 and highlighted in green in **i** for ER, and **j** for Golgi. Nuclei (Nu) of the cells are stained by Hoechst 33342 and shown in cyan. The brightness and contrast of the images were digitally adjusted to enhance signal visibility. Scale bar, 10 µm. **k** Total expression of SBP-E2-7TM without and with GPS mutations with hook protein, without biotin addition, was measured by total ELISA. Data are expressed as the percentage of the signals from SBP-E2$^{WT}$-7TM+hook. All data points were plotted and represented as box-and-whisker diagrams with parameters as stated in Fig. 1c. The *p*-values for each comparison are listed. **l** The surface expressions of SBP-E2-7TM with or without GPS mutations before biotin addition were measured by surface ELISA. Data are expressed as the percentage of the signals from SBP-E2$^{WT}$-7TM. Data were extracted from **m**. All data points were plotted and represented as box-and-whisker diagrams with parameters as stated in Fig. 1c. The *p*-values for each comparison are listed. **m-n** The surface expressions of SBP-E2-7TM with or without GPS mutations after the indicated duration of biotin treatment were measured by surface ELISA. Data are expressed as **m** the percentage of the signals from SBP-E2$^{WT}$-7TM after 6 hours of biotin addition, or **n** the fold of the initial signals of the variants. Data are shown as mean ± SEM indicated in shaded area. The yellow rectangle indicates a significant decrease ($p < 0.008$) in the surface expressions of the GPS mutants of SBP-E2-7TM than the WT. The comparison on the surface expression at $t = 0$ is shown in (**l**). **o** GAIN domain cleavage of E2 serves as a checkpoint to permit ER exit of the receptor, leading to receptor trafficking. Created in BioRender. Chung, Y. (2025) https://BioRender.com/hstefm2. Source data are provided as a Source Data file. See Methods section for statistics and reproducibility.

## ER membrane proximity is critical for GAIN domain cleavage of various aGPCRs

As the attachment of E2-ECR-CalnTM$^{ER}$ to the ER membrane promoted its autoproteolysis, proximity of the GAIN domain to the membrane may contribute to this effect (Fig. 1l). To test this hypothesis, we inserted an mEmerald spacer between ECR and CalnTM (E2-ECR-Fluo-CalnTM$^{ER}$) adding a maximum of 50 nm distance (Fig. 4a)[35–37]. Remarkably, this caused a large reduction in E2 GAIN domain cleavage efficiency from 45% to 10 % (Fig. 4b, c). This was also observed in a series of autoproteolysable aGPCR-ECRs from different subfamilies (E, G, L, D) and species (rat, *Drosophila*) we tested (Fig. 4c, d and Supplementary Fig. 5), indicating that ER membrane proximity enhances GAIN domain cleavage efficiency across the aGPCR family.

We next gradually distanced the GAIN domain from the lumenal ER membrane side via insertions of a variable number of Glu-Ala-Ala-Ala-Lys helical turns (#ht; # indicates the number of turns) into E2-ECR-CalnTM$^{ER}$ (E2-ECR-#ht-CalnTM$^{ER}$). With increasing ht number, GAIN domain cleavage decreased (Fig. 4e), corroborating the ER membrane proximity effect observed on E2$^{WT}$-ECR-Fluo-CalnTM$^{ER}$ processing (Fig. 4b). The proximity-cleavage function was inversely exponential ($R^2 = 0.9234$) rather than biphasic (Fig. 4f), implying that membrane proximity impacts GAIN domain cleavage by increasing the probability for effective proteolysis. The extent E2$^{WT}$-ECR-Fluo-CalnTM$^{ER}$ cleavability was similar to the one of E2-ECR-#ht-CalnTM$^{ER}$ containing 4 or more ht (adding ca. 20 nm) (Fig. 4f)[35,36]. These results collectively show that membrane proximity assists in GAIN domain autoproteolysis (Fig. 4g).

## A proteomic approach to identify ER- and GAIN domain-restricted protein interactions of aGPCRs

A logical mechanism by which ER membrane proximity of nascent aGPCR proteins aids in GAIN domain cleavage is the localisation of the domain in the vicinity of ER membrane-associated co-factors. For example, components of the translocon system (Sec61, Sec63) and the unfolded protein response (UPR) pathway component Xbp1 are required for GAIN domain cleavage of PC1, E2 and L1[38]. To this end, we used an unbiased proteomic approach to systematically explore co-factors for GAIN domain autoproteolysis residing at or in the ER membrane.

We overexpressed and pulled down the ER-localised E2-ECR proteins E2$^{WT}$-ECR-CalnTM$^{ER}$, E2-NTF-KDEL$^{ER}$, E2$^{WT}$-ECR-10ht-CalnTM$^{ER}$, and E2$^{H516A}$-ECR-CalnTM$^{ER}$ (Figs. 1h and 4e), to determine their interactors with mass spectrometry (MS) without crosslinking. The E2-NTF-KDEL$^{ER}$ control is unlikely to form a folded GAIN domain because the last β-sheet of the GAIN domain is deleted[20], and additionally, it is not tethered to the ER membrane. In contrast, E2$^{WT}$-ECR-10ht-CalnTM$^{ER}$ and E2$^{H516A}$-ECR-CalnTM$^{ER}$ possess a complete GAIN domain and are tethered to the ER membrane, but the former protein is distanced from the ER membrane, whereas the latter one is not but possessed cleavage deficiency. Therefore, comparisons on the likelihood of binding of proteins between E2$^{WT}$-ECR-CalnTM$^{ER}$ and the other three protein designs provided information on the possible co-factors assisting GAIN domain proteolysis.

Compared to E2$^{WT}$-ECR-CalnTM$^{ER}$, 51 and 45 proteins were differentially co-immunoprecipitated by E2-NTF-KDEL$^{ER}$ and E2$^{WT}$-ECR-10ht-CalnTM$^{ER}$, respectively (Fig. 5a, b, Supplementary Table 2). Of these sets, 25 and 10 proteins, respectively, are predominantly ER localised (Fig. 5a, b, Supplementary Table 2), implying that interactions with these proteins may be sensitive to the proper conformation of the GAIN domain (in the case of E2-NTF-KDEL$^{ER}$), or the proximity to the GAIN domain (in the case of E2$^{WT}$-ECR-10ht-CalnTM$^{ER}$). In contrast, no proteins were differentially pulled down between E2$^{WT}$-ECR-CalnTM$^{ER}$ and E2$^{H516A}$-ECR-CalnTM$^{ER}$, suggesting that the overall conformation of cleaved and mutationally uncleavable GAIN domains and thus their

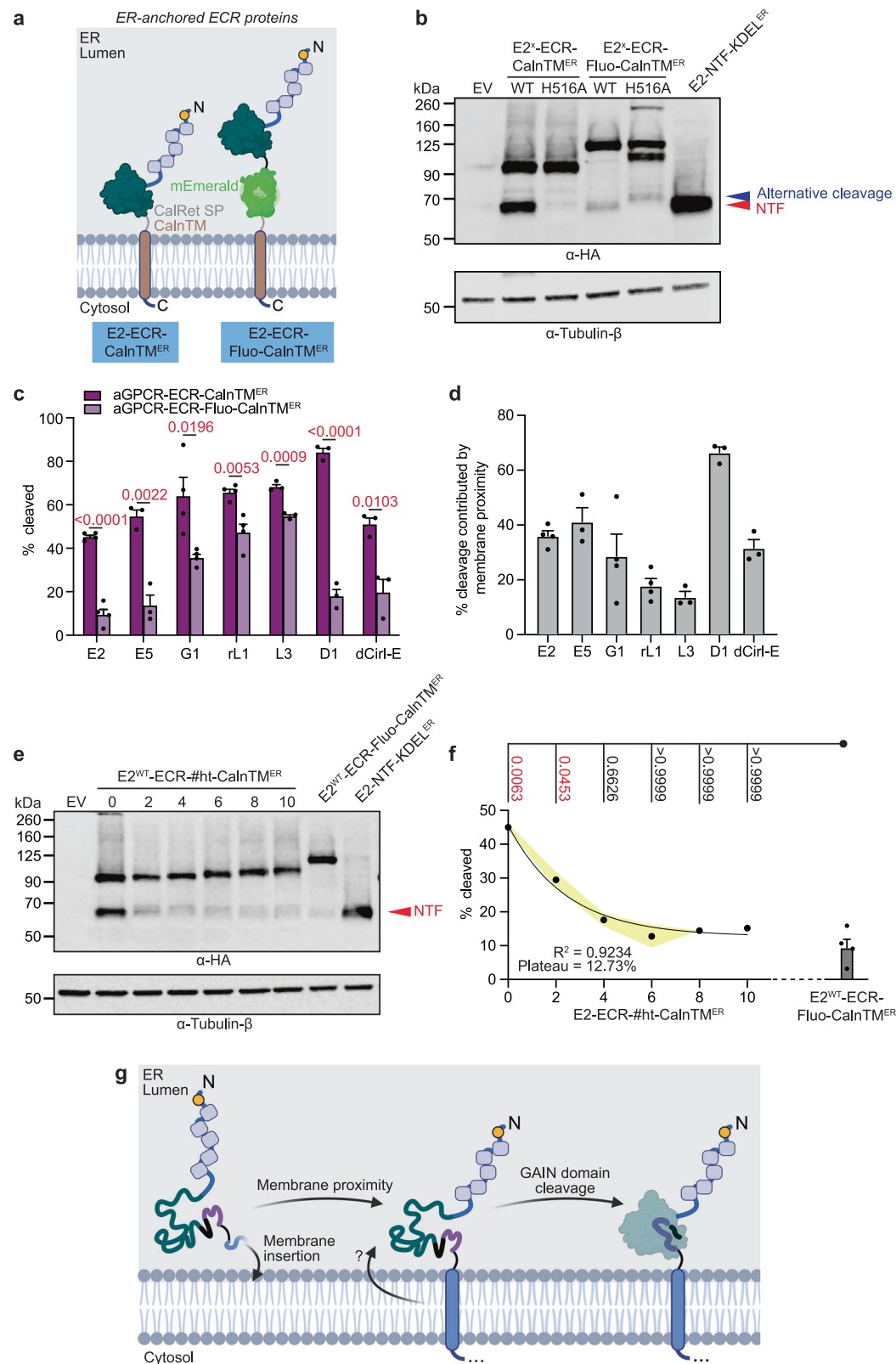

binding interfaces are similar (Fig. 5c, Supplementary Table 2). This dataset comes with two limitations. First, the interaction between the putative cofactors and the E2 protein designs are not necessarily specific to the GAIN domain. Second, the presence of the full 7TM domain may require different modulators for GAIN domain cleavage than single TM-containing variants.

Therefore, we extended our screen for folding or cleavage assisting interactors with an assay to focus on ER-resident membrane-proximal interactors of an E2-7TM variant via site-specific photo-crosslinking. To this end, we used Genetic Code Expansion (GCE)[7,39–42] to individually replace the five phenylalanine residues of the GAIN domain with p-benzoyl-L-phenylalanine (Bpa), a photo-activatable

**Fig. 4 | The proximity to the luminal side of the GAIN domain impacts proteolysis of aGPCRs. a** Design of the ECR proteins that tether the ECR of E2 to the lumenal side of the ER membrane. The insertion of mEmerald, as in ECR-Fluo-CalnTM$^{ER}$, acts as a domain spacer between the lumenal side of the ER membrane and ECR. Created in BioRender. Chung, Y. (2025) https://BioRender.com/u0vmadl. **b** Representative Western blot showing the expression patterns of the ER-anchored E2-ECR proteins, detected against the N-terminal HA tag. Bands representing NTF and alternative cleavage are labelled in red and deep blue respectively. Detection of tubulin-β is also shown as a loading control. Quantification of **c** the extent of GAIN domain cleavages, and **d** the percentage of GAIN domain cleavage of ER-anchored ECRs of different aGPCRs contributed by membrane proximity. Data are shown as mean ± SEM. Individual values are plotted. The *p*-values for each comparison are listed, and those below the confidence interval (95%) are labelled in red. The representative Western blots are shown in Supplementary Fig. 5. **e** Representative

Western blot showing the expressions of E2$^{WT}$-ECR-CalnTM$^{ER}$ inserted with various helical turns (#ht), detected against the N-terminal HA tag. Detection of tubulin-β is also shown as a loading control. **f** Quantification of the extent of GAIN domain cleavage of E2$^{WT}$-ECR-#ht-CalnTM$^{ER}$. Data are shown as mean ± SEM shown as shaded area. The *p*-values for each comparison are listed, and those below the confidence interval (95 %) are labelled in red. The trend was approximated into a first-order decay. The coefficient of determination ($R^2$), and the plateau cleavage percentage are also shown. The extent of GAIN domain cleavage became similar to E2$^{WT}$-ECR-Fluo-CalnTM$^{ER}$ for E2$^{WT}$-ECR-#ht-CalnTM$^{ER}$ constructs with 4 and longer helical insertions (indicated in *p*-values labelled in red). **g** The insertion of the ECR of E2 upon the synthesis of TM1 provides a proximity of the GAIN domain to an unknown factor, which assists the GAIN domain proteolysis. Created in BioRender. Chung, Y. (2025) https://BioRender.com/oarp85p. Source data are provided as a Source Data file. See Methods section for statistics and reproducibility.

unnatural amino acid (UAA) (Supplementary Fig. 6a). These residues are at positions encoding the solvent-exposed surface of the GAIN domain in SBP-E2-7TM transgenes (Supplementary Fig. 6a, b). Targeted amino acid replacement by GCE converts the respective phenylalanine-encoding codons into an amber stop codon (SBP-E2$^{Phe>*}$–7TM) (Supplementary Fig. 6a, b). Co-expression of SBP-E2$^{Phe>*}$–7TM transgenes with a Bpa-specific amino-acyl-tRNA-synthetase (BpaRS) allows the incorporation of Bpa (provided in the culture medium) into the nascent receptor protein chain as a bioorthogonal single-amino acid label (Supplementary Fig. 6a, b). We also co-expressed the bioorthogonally tagged SBP-E2$^{Phe>Bpa}$-7TM proteins with the ER-localised hook of the RUSH assay (Fig. 3H) to capture the ER-resident candidates (Supplementary Fig. 6a, b, lanes 'B' for Bpa)[39,43]. Subsequent UV irradiation crosslinked local interacting proteins to the E2 GAIN domain (Supplementary Fig. 6a, b, lanes 'BU' for Bpa + UV irradiation). As a control, we inserted Bpa into the second EGF domain of E2 and achieved similar photo-crosslinking, suggesting the applicability of the approach to other regions of the receptor protein (Supplementary Fig. 6b).

The highest crosslinking yield was observed in SBP-E2$^{F355Bpa}$-7TM and SBP-E2$^{F469Bpa}$-7TM (Supplementary Fig. 6b), while SBP-E2$^{F467Bpa}$-7TM and SBP-E2$^{F469Bpa}$-7TM showed impeded GAIN domain cleavage (Supplementary Fig. 6b). To confirm that the synthesised photosensitive SBP-E2$^{Phe>Bpa}$-7TM mutants were held in the ER, we analyzed surface expression of the receptors by ELISA. Only after biotin treatment did a significant amount of the respective SBP-E2$^{Phe>Bpa}$-7TM proteins reach the membrane (Supplementary Fig. 6c), demonstrating the successful combination of RUSH and GCE. Consequently, SBP-E2$^{Phe>Bpa}$-7TM receptors can be captured within the ER to assess their local proteome.

### N-glycosylation contributes to GAIN domain cleavage of E2

We chose SBP-E2$^{F355Bpa}$-7TM for further MS-based proteome analysis. A total of 53 proteins were identified via photo-crosslinking with SBP-E2$^{F355Bpa}$-7TM, including the glucosidase IIα subunit (GIIα) (encoded by *GANAB*) as a prominent ER-based interactor of SBP-E2$^{F355Bpa}$-7TM (Fig. 5d, Supplementary Table 3). We ranked potential ER-resident interaction partners identified in photo-crosslinking and non-crosslinking experiments by pathway analysis. ER-located protein binding partners were clustered into two groups involved in protein N-glycosylation, cellular responses to misfolded proteins (e.g., ER stress/ER-associated degradation, ERAD/UPR pathways), and glycosylphosphatidylinositol (GPI) anchorage (Fig. 5e). Gene Ontology (GO) enrichment analysis (Supplementary Table 4) and KEGG Pathway Database (Supplemetary Table 5) confirmed allocation to these pathways.

We focused on the effect of N-glycosylation on the GAIN domain cleavage of E2, a pathway which was previously reported to affect aGPCR processing and trafficking (Fig. 5f and Supplementary Fig. 7)[44]. We observed that all candidates exclusively involved in N-glycosylation

have an increased pulldown upon membrane tethering, while proteins that also participate in the redirection of proteins towards ER stress-related cascades show a decreased interaction (Supplementary Fig. 7). These results support the model that N-glycosylation plays an important role in GAIN domain cleavage of E2, possibly via regulation of protein folding.

To test this conjecture, cells overexpressing E2$^{WT}$-ECR-CalnTM$^{ER}$ were pharmacologically treated to inhibit N-glycan synthesis by tunicamycin (Tun), SST3A and SST3B (catalytic subunits of the oligosaccharyltransferase (OST) complex) activity by NGI-1, and GIIα activity by castanospermine (Castano), comprising several key steps of the protein N-glycosylation cascade (Fig. 5f, g). All three drugs effectively impaired N-glycosylation as tested by protein mobility shift assays after PNGase F treatment (Fig. 5g). Accordingly, E2$^{WT}$-ECR-CalnTM$^{ER}$ of Tun-treated cells was devoid of N-glycans, as PNGase F treatment did not shift the mobility of the FL protein (Fig. 5g, lanes '-'). E2$^{WT}$-ECR-CalnTM$^{ER}$ in cells treated with NGI-1 showed a reduction of protein size without PNGase F treatment, indicating a decrease in N-glycosylation (Fig. 5g, lanes '-'). This suggests that E2 could undergo N-glycosylation by an OST-independent pathway. E2$^{WT}$-ECR-CalnTM$^{ER}$ displayed an increased mass upon Castano treatment (Fig. 5g, lanes '-'). The mobility shifts caused by the inhibitors are due to changes in N-glycosylation rather than the protein itself as concluded by PNGase F addition to E2$^{WT}$-ECR-CalnTM$^{ER}$ bands, which resolved to the same size and were similar to those in the mock-treated condition (Fig. 5g, lanes 'P').

Finally, quantification of E2$^{WT}$-ECR-CalnTM$^{ER}$ cleavage showed that the GAIN domain autoproteolysis is largely disrupted only upon Tun treatment, when no N-glycans are added to the receptor (Fig. 5h). This shows that the N-glycosylation of the E2-ECR is required for GAIN domain cleavage. Interestingly, while the defect of proteolysis by complete abolishment of N-glycosylation (via Tun treatment) is apparent for E2$^{WT}$-ECR-CalnTM$^{ER}$ (Fig. 5g, h), E2$^{WT}$-3TM (Supplementary Fig. 8a) and E2$^{WT}$-5TM (Supplementary Fig. 8b), it is potently rescued in presence of the entire 7TM domain (Fig. 5I and Supplementary Fig. 8c). This suggests that the effect of N-glycosylation of the GAIN domain is exerted before the completion of 7TM biogenesis (Fig. 5j). Moreover, Tun treatment suppressed surface expression of E2$^{WT}$-xTMs, while all pharmacological inhibitions reduced the surface delivery rates of SBP-E2-7TM GPS variants in response to biotin treatment in the RUSH assay (Supplementary Fig. 9a, b), indicating that the N-glycosylation of E2 contributes to the ability of the receptor to enter the trafficking route.

### Discussion

Previous studies have established that the isolated GAIN domain is sufficient and necessary for receptor proteolysis at the GPS[13,20]. However, several observations implying that GAIN domain-mediated autoproteolysis of self-cleavable aGPCRs is context-dependent contrast with this sentiment, as cell type, developmental stage, additional extracellular receptor domains, receptor concentration, and post-

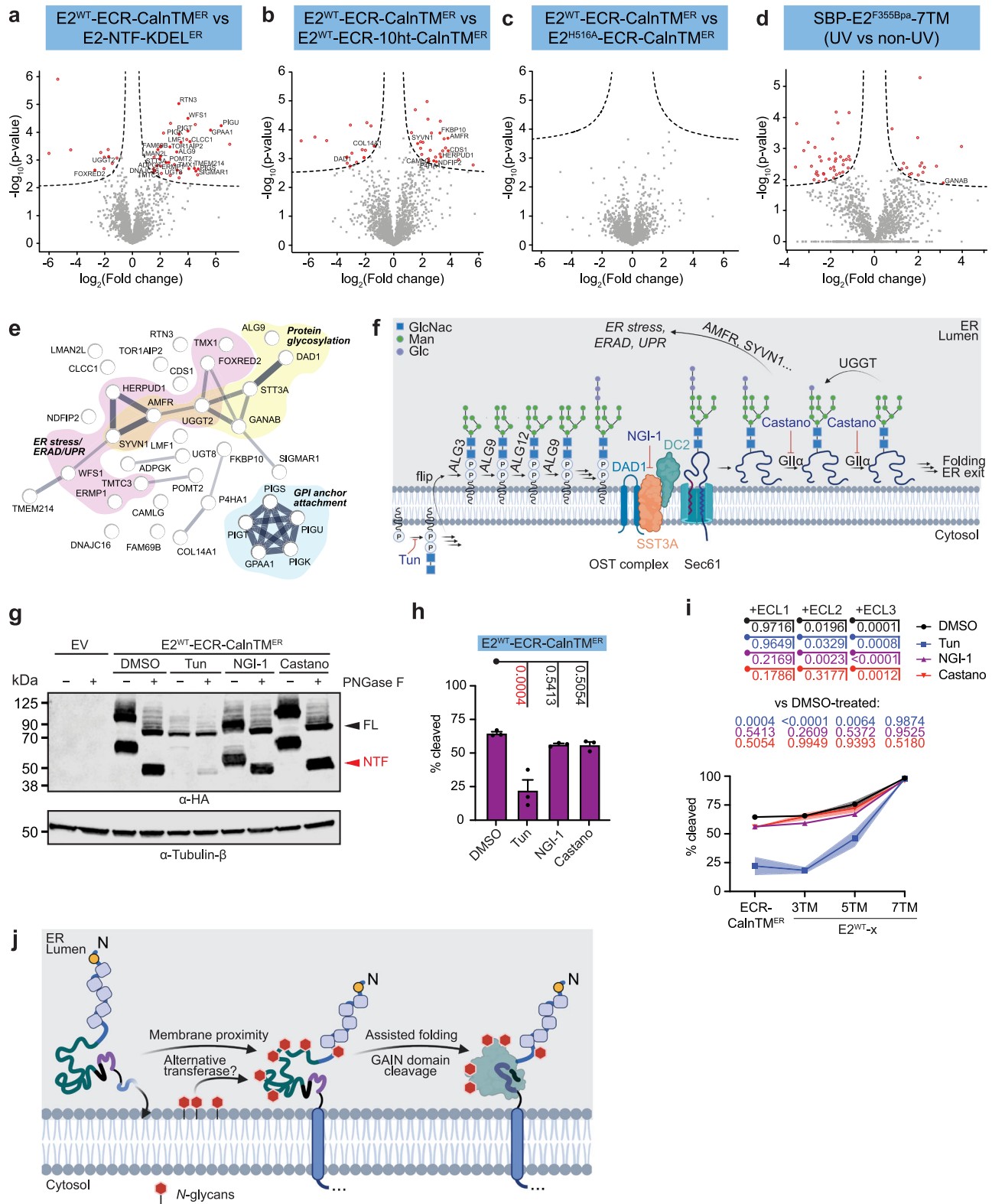

translational modifications affect GAIN domain cleavage. For example, ADGRB3/BAI3 showed autoproteolytic processing when isolated from brain tissue, while when expressed in cell culture, the receptor remained uncleaved[4]. Further, autoproteolytic activity of ADGRG1/GPR56 declined during the progress of brain development[17]. GAIN domain cleavage of a specific ADGRE5/CD97 isoform expressed as an ADGRE5ECR-Fc fusion upscaled with protein concentration, leading the authors to even suggest an intermolecular rather than an

intramolecular reaction[18]. In addition, this study also proposed that folding and expression of the GAIN domain depend on the presence of an adjacent EGF domain in the ECR of the receptor, and that N-glycosylation of ADGRE5 aids GAIN domain cleavage[18], corroborating earlier findings with ADGRE5-Fc fusion proteins[45]. Disease conditions may also interfere with or even act through changes in GAIN domain cleavage via any of the above mechanisms, as exemplified by the loss of self-cleavage of ADGRE5 in rhabdomyosarcomas[19]. Based on these

**Fig. 5 | *N*-glycosylation of the GAIN domain of E2 aids in the cleavage event. a–d** Volcano plots of the proteins (**a–c**) co-immunoprecipitated with ER-E2-NTF[ER], E2[WT]-ECR-CalnTM[ER], E2[H516A]-ECR-CalnTM[ER] and E2[WT]-ECR-10ht-CalnTM[ER], and (**d**) photo-crosslinked with SBP-E2[F355Bpa]–7TM by shotgun proteomic approach. Proteins that are significantly different in the likelihood of pulldown between **a** E2[WT]-ECR-CalnTM[ER] and ER-E2-NTF[ER], **b** E2[WT]-ECR-CalnTM[ER] and E2[WT]-ECR-10ht-CalnTM[ER], and **c** E2[WT]-ECR-CalnTM[ER] and E2[H516A]-ECR-CalnTM[ER], or **d** significantly different in the likelihood of photo-crosslinking upon UV irradiation are shown in red circles. ER-localised proteins are filled in red and labelled in cyan, while the non-ER-localised proteins are represented as red open circles. Lists of identified proteins are shown in Supplementary Table 2-3. *N* = 3. **e** Pathway analysis on the identified ER-localised proteins. Proteins are clustered into the relevance towards protein glycosylation (yellow), ER stress/ERAD/UPR pathways (magenta) and GPI anchor attachment (blue). Thickness of the connecting lines indicate the level of functional association. Lists of pathways identified are shown in Supplementary Table 4-5. **f** Schematics of protein *N*-glycosylation. The proteins in the cluster relevant to protein glycosylation in (**e**) are listed. The LFQ intensity changes of the proteins obtained from the MS analyses are shown in Supplementary Fig. 7. Small molecule inhibitors of several stages of the *N*-glycosylation process, used in subsequent assays, are also shown. Created in BioRender. Chung, Y. (2025) https://BioRender.com/telxz8s. **g, h** HEK293T cells were transiently expressed with E2[WT]-ECR-

CalnTM[ER] for 24 hours, before treatments with Tun (10 ng/mL), NGI-1 (10 μM) or Castano (100 μg/mL) for an additional 24 hours before lysis. Receptors with or without PNGase F treatment were verified by Western blotting against the N-terminal HA tag. Uncleaved product is indicated in black triangles, while cleaved products are shown in red triangles. Detection of tubulin-β is also shown as a loading control. Quantification on the extent of GAIN domain cleavage is shown in (**h**) and also (**i**). Data are shown as mean ± SEM. The *p*-values for each comparison are listed, and those lower than the level of significance are labelled in red. **i** Cells expressing E2[WT]-ECR-CalnTM[ER], E2[WT]-3TM, E2[WT]-5TM or E2[WT]-7TM were subjected to overnight pharmacological inhibition against different stages of *N*-glycosylation. The extent of cleavage was quantified. Quantification concerning E2[WT]-ECR-CalnTM[ER] is also shown in (**h**). Representative blots are shown in Supplementary Fig. 8a–c. Data are shown as mean ± SEM shown as a shaded area. The *p*-values for each comparison are listed following the colour codes of the mutants. **j** The membrane proximity allows the attachment of the *N*-glycans to the ECR of E2, including on the GAIN domain, which contributes to GAIN domain proteolysis presumably by the assistance in correct folding of the domain. The transfer of the *N*-glycans is seemingly independent of the action of STT3A/STT3B. Created in BioRender. Chung, Y. (2025) https://BioRender.com/zcc2nfj. Source data are provided as a Source Data file. See Methods section for statistics and reproducibility.

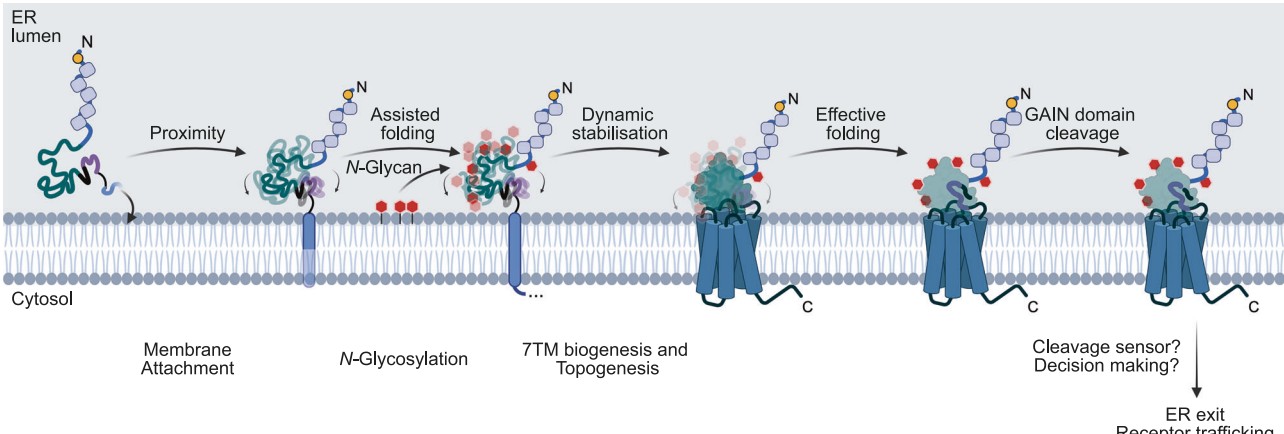

**Fig. 6 | GAIN domain cleavage of E2 requires multiple domain-extrinsic factors coordinated by the 7TM region.** The generation of the TM1 anchors the GAIN domain onto the luminal side of the ER membrane, which increases the proximity for the transfer of *N*-glycans. The *N*-glycosylated domain folds more properly, albeit still highly dynamic, disfavouring the proteolysis. It is resolved by the gradual

synthesis of the entire 7TM region, which stabilises the dynamics of the GAIN domain and strongly promotes effective GAIN domain cleavage. This ultimately leads to the permission of the receptor to enter the trafficking route via an unknown mechanism. Created in BioRender. Chung, Y. (2025) https://BioRender.com/hhtsnft.

reports and the large number of self-cleavable aGPCRs, we systematically analysed the impact of the 7TM regions in modulating the efficiency of GAIN domain proteolysis, and their effect on fate and function of the receptor protein, using ADGRE2 as a model receptor.

We determined an important role of the receptor 7TM region in GAIN domain cleavage of newly synthesised receptor protein (Fig. 6). These concern two aspects: the tethering of the GAIN domain to the lumenal side of the ER membrane (by TM1), which subsequently promotes proximity of the GAIN fold to the *N*-glycosylation machinery (Figs. 1, 4, 5), and the gradual synthesis of the entire 7TM region (Fig. 2). All contributions share the same aim to fold and stabilise the highly dynamic GAIN domain[7,31], which presumably allows effective initiation of the proteolysis event.

When receptor synthesis proceeds after generation of the GAIN domain, the nascent TM1 helix anchors the GAIN domain onto the lumenal side of the ER membrane upon its membrane insertion. This process is critical for GAIN domain cleavage, possibly via facilitating domain folding (Fig. 1i-l). This relationship is in agreement with earlier findings in L1[46], in which only the membrane-bound but not the soluble version of the GAIN domain can interact with its binding partner, α-

latrotoxin, with high affinity. It suggests that membrane tethering is required for the proper folding of the GAIN domain, which subsequently aids GAIN domain cleavage[20,46]. Interaction of GAIN and 7TM domains may then restrict structural flexibility of the GAIN domain[7,31]. This aligns with structural analyses on the 7TM-bound ECR of ADGRL3[24], where the GAIN domain is positioned in parallel to the membrane surface. Further, single-molecule FRET analyses suggested that GAIN domain conformations are limited to three states with low exchange rates between different conformations[24].

We further characterised the importance of membrane proximity of the GAIN domains of all tested aGPCRs (Fig. 4), and identified, by a GAIN domain-centric proteomic approach, that this benefits *N*-glycosylation (Fig. 5). Previous reports have described *N*-glycosylation of several aGPCRs, including E5 and D1[28,45], and glycosylation of E2 was suggested to promote plasma membrane anchoring of shed NTF[44]. 7TM-bound cryo-EM structure of the GAIN domain of ADGRL3 detected *N*-glycans at the membrane-facing surface of the domain[24], supporting our findings that membrane tethering via TM1 insertion positions the GAIN domain in the vicinity of the ER-resident *N*-glycosylation machinery to facilitate protein folding.

Our study sheds light on the importance of GAIN domain cleavage for the subcellular trafficking of aGPCRs exemplified by the immune receptor ADGRE2/EMR2 (Fig. 3e-n). Although GPS mutations did not alter the steady-state surface and total expressions of E2 (Fig. 3e, f), we were able to show that the incapability of GAIN domain cleavage decreased the quantity of receptors to exit from the ER without affecting the rate of surface delivery (Fig. 3i-n). This was performed using a RUSH approach, which was adapted to study aGPCRs and is now available to future studies on the domain-specific functions and modifications of aGPCRs.

## Methods

### Plasmids
A list of all plasmids used in the study, and the primers used for generations, is explained in Supplementary Table 6 and Supplementary Table 7, respectively.

### Antibodies
Rabbit α-HA was purchased from Cell Signalling Technology (C29F4, Cat. No. 3724). Rat α-HA-Peroxidase High Affinity (α-HA-HRP) was obtained from Roche (Cat. No. 12013819001). Mouse α-RFP was provided by Chromotek (Cat. No. 6G6). Mouse α-tubulin-β was obtained from DSHB (E7). Alexa Fluor™ 647-conjugated α-HA was purchased from Invitrogen (Cat. No. 26183-A647). IRDye 680RD Goat anti-Rabbit (Cat. No. 926-68071) and IRDye 800CW Goat anti-Mouse (Cat. No. 926-32210) were purchased from Li-Cor.

### Chemicals
*PfuUltra* High-Fedelity DNA polymerase was obtained from Agilent (Cat. No. 600382). T4 ligase (Cat. No. EL0011), Pierce™ anti-HA magnetic beads (Cat. No. 88837), Pierce™ IP lysis buffer (Cat. No. 87787), and Halt™ Protease inhibitor cocktail (100×) (Cat. No. 78429) were purchased from Thermofisher. Other enzymes used in plasmid constructions, and *N*-glycosidases were obtained from New England Biolabs (NEB). The following materials were provided by Sigma: DMEM high-glucose (Cat. No. D6429), Poly-L-lysine (Cat. No. P9404), hydrogen peroxide (Cat. No. 216763), o-Phenylenediamine (Cat. No. P9029), tunicamycin (Cat. No. T7765) and D-Biotin (Cat. No. B4501). Heat-inactivated FBS (Cat. No. 10500064), penicillin/streptomycin (Cat. No. 15140122), StemPro™ Accutase™ Cell Dissociation Reagent (Cat. No. A1110501), $CO_2$-independent medium (Cat. No. 18045088), and DPBS(-) (Cat No. 14190144) were purchased in Gibco. Live Cell Imaging Solution (LCIS) (Cat. No. A14291DJ) and Lipofectamine™ 2000 transfection reagent (Cat. No. 11668027) were supplied from Invitrogen. Castanospermine (Cat. No. HY-N2022) and NGI-1 (Cat. No. HY-117383) were obtained from MedChemExpress. Other chemicals were purchased from either Carl Roth or Sigma, unless otherwise specified.

### Cell culture and transfection
HEK293T cells were obtained from ATCC (cat. No. CRL-3216). Cells were maintained in DMEM high-glucose supplemented with 10% heat-inactivated FBS and 1% penicillin/streptomycin, at 37 °C, 5% $CO_2$ and saturated humidity. Cells for experiments were below 50 passages.

Cells were seeded in the designated format one day before transfection to reach a confluency of 70-80% on the next day. Plates were coated with 1% (w/v) poly-L-lysine for 45 minutes under a benchtop UV lamp before seeding. Except for the RUSH assay, cells were transfected by Lipofectamine™ 2000 transfection reagent and DMEM high-glucose following the manufacturer's instructions. Per 1000 ng of plasmid, 4 μl of transfection reagent was used. Cells were incubated for 24 hours before downstream analyses, except experiments involving *N*-glycosylation inhibitors in which cells were analysed after 48 hours, with an overnight drug treatment 24 hours post-transfection.

Transfection for the RUSH assay was performed by the calcium phosphate method[34]. Per well of a 96-well, 160 ng of plasmid was mixed with 3.4 μl of 340 mM $CaCl_2$, followed by 5 μl of 2× Hank's Buffer Saline Solution (289 mM NaCl, 2.8 mM $Na_2HPO_4$, 50 mM HEPES, pH 7.2). The mix was immediately applied to a well of cells that were refreshed by 90 μl full DMEM. Cells were incubated for a total of 24 hours before ELISA detection. Cells were directly added with Biotin (final 40 μM) at required time points prior to ELISA.

### Western Blotting
Cells were seeded in a 6-well plate or 24-well plate (Greiner, cat. No. 657160 and 662160, respectively) one day before transfection. 1600 ng (for 6-well) or 400 ng (for 24-well) of plasmids were transfected per well. After 24 hours, transfected cells were lysed in Lysis Buffer (50 mM Tris pH 8.0, 150 mM NaCl, 1% Triton X-100) supplemented with protease inhibitors. Lysates were precleared of insoluble debris by centrifugation at 4 °C, 10,000× $g$ for 10 minutes. Unless otherwise specified, lysates were then applied with self-made 5× Laemmli buffer. After brief vortexing, samples were loaded onto 4-20% Novex™ Tris-Glycine Mini Protein Gels (Invitrogen, cat. No. XP04200BOX or XP04202BOX). Thirty μl of the samples were loaded in each well. After the dye front has reached the bottom of the gel, the gels were excised, and proteins were transferred onto nitrocellulose membranes using iBlot™ 2 Transfer System (Invitrogen). The membranes were then washed twice in PBS-0.1% Tween-20 (PBST) and blocked by blocking buffer (0.1% (v/v) Tween-20 in 1:1 PBS and Intercept® Blocking Buffer obtained from Li-Cor, cat. No. 927-70001) for 1 hour at room temperature. Membranes were incubated in blocking buffer-diluted primary antibodies overnight at 4 °C with shaking, using the following dilution: α-HA, 1:1000; α-RFP, 1:1000; α-tubulin-β, 1:5000. After that, the membranes were washed twice with PBST, followed by incubation with secondary antibodies in dilution 1:15,000 in PBST, for 1 hour at room temperature with shaking. Membranes were then washed twice with PBS. Antibody signals were detected using Odyssey® XF Dual-Mode Imaging System (Li-Cor).

### Deglycosylation of proteins
26 μl of lysates were mixed with or without 1 μl of PNGase F or 1 μl of Endo H supplemented with the appropriate buffer following the manufacturer's protocol. The mixture was incubated for 1 hour at room temperature, and was terminated by the addition of 5×Laemmli buffer.

### Enzyme-linked immunosorbent assay (ELISA)
Cells were seeded in a 96-well plate (Greiner, cat. No. 655087) one day before transfection. 100 ng of plasmids were transfected per well. Each construct was tested in a quadruplicate format in each trial. After the desirable time, each well of transfected cells was fixed in 50 μl of 4% paraformaldehyde for 10 minutes at room temperature. Cells were then blocked with 100 μl of 5% FBS in PBS (for surface expression detection) or PBST (for total expression detection) for 30 minutes at room temperature. After that, cells were applied with 100 μl of antibody solution (α-HA-HRP, 1:1000, in PBS containing 5% FBS) for 1 hour at room temperature. Cells were washed twice with 200 μl of PBS. HRP reaction was started by adding 100 μl of substrate solution (1 μl/mL $H_2O_2$ and 1 mg/mL o-Phenylenediamine in pH 5.0 ELISA buffer consisting of 0.05 M citric acid and 0.05 M $Na_2HPO_4$) for at most 10 minutes. Reaction was terminated by adding 100 μl of 2.5 M sulphuric acid. 150 μl of the solution was transferred to an empty 96-well plate. Absorbance at 490 nm was detected using SpectraMax iD5 microplate reader (Molecular Device). Signals contributed by the expressed receptor were calculated by subtracting the raw absorbance from the mean absorbance of cells transfected with the empty vector.

## Confocal microscopy

Cells were seeded in an 8-well chamber slide (Ibidi, cat. No. 80841) one day before transfection. For the RUSH assay, 480 ng of plasmids were transfected per well by the calcium phosphate method. After 24 hours, cells were washed twice with PBS and then kept in $CO_2$-independent medium with or without 40 μM of biotin. For the visualisations of E2 mutants on their subcellular localisations, 200 ng of plasmids were transfected per well by Lipofectamine™ 2000. One day after transfection, cells were fixed in 4% (w/v) PFA and permeabilised with 5% FBS in PBST. Cells were then incubated with Alexa-Fluor™−647-conjugated α-HA (1:500) for 30 minutes. Cells were washed twice with PBS and kept in PBS for imaging. Fluorescence of eGFP (for RUSH assay) or Alexa-Fluor 647 was monitored using Leica SP8 microscope (63×/1.3 glycerol objective). In all experiments, ER or Golgi were visualised by CellLight ER-RFP or Golgi-RFP BacMam 2.0 technology (Thermofisher, Cat. No. C10591 and C10593, respectively). Nuclei were stained by Hoechst 33342 (Thermofisher, Cat. No. 62249). Images were processed by ImageJ. The brightness and contrast of the images were digitally adjusted to enhance signal visibility.

## MS sample preparation

For non-crosslinked E2 ECR protein sample preparations, cells were transfected in a 10cm-plate format with 10 μg of plasmids by the Lipofectamine™ 2000 method. For sample preparations involving genetic code expansion using Bpa and photo-crosslinking of cells, it was performed similarly to Böttke et al[39]. In short, cells seeded in a 10cm-plate format were cotransfected with 5 μg each of a bicistronic vector expressing the ER-localised hook and the amber stop codon-incorporated SBP-E2$^{Phe>*}$-7TM, and the expression vector for Bpa-tRNA synthetase. Cells were incubated in media containing Bpa (final concentration of 250 μM) 1 hour prior to transfection. One day after transfection, cells were washed twice with ice-cold DPBS. Then, cells kept in ice-cold DPBS were irradiated with UV-A light (365 nm) for 15 minutes perpendicularly to a benchtop UV lamp. As control samples, cells were not irradiated with UV-A light. After that, DPBS was removed and the cells were frozen in −80 °C for 30-60 minutes before lysis.

Cells were lysed in 1 ml of lysis buffer (50 mM Tris, pH 8.0, 150 mM NaCl, 1% Triton X-100) supplemented with protease inhibitors, pre-cleared, and subjected to immunoprecipitation using the N-terminal HA tag with Pierce™ Anti-HA Magnetic Beads following the manufacturer's protocol, except that washing by TBST was replaced by Pierce™ IP buffer. Beads were snap-frozen in 100 μl milli-Q water and kept in −80 °C until further analyses.

Samples were prepared for LC-MS/MS analysis following a modified filter-aided sample preparation (FASP) protocol[47]. Briefly, protein samples were incubated and washed (14,000× g for 2 × 10 min) with 8 M urea in 50 mM HEPES, pH 8.5, 10 mM TCEP in 0.5 mL centrifugal filter units (30-kDa cutoff) (Sigma Aldrich, Darmstadt, Germany). Samples were then alkylated with 50 mM iodoacetamide in 8 M urea, 50 mM HEPES (pH 8.5) at room temperature for 20 min in the dark and washed twice with 50 mM HEPES, pH 8.5 (2 × 10 min at 18,000 × g) before incubation with 0.5 μg trypsin (Promega) in 50 mM HEPES (pH 8.5) at 37 °C overnight. After digestion, samples were acidified with TFA (final concentration 0.5 % (v/v)). A total of 4 samples (for non-crosslinked E2 ECR proteins) and 2 samples (for Bpa-mediated photo-crosslinking) were analysed per experiment, and 3 experimental replicates were performed.

## LC-MS/MS analyses

Peptide solutions were analyzed by LC-MS/MS on an Ultimate 3000 RSLC nano-HPLC system (Thermo Fisher Scientific) coupled to a CaptiveSpray ion source of a timsTOF Pro mass spectrometer (Bruker Daltonik). Peptides were trapped on a reversed-phase C18 precolumn (Acclaim PepMap 100, 300 μm × 5 mm, 5 μm, 100 Å, Thermo Fisher Scientific) and washed for 15 min 0.1% (v/v) TFA in water (flow rate

30 μL/min, temperature 50 °C). Using a constant flow of 300 nL/min, peptide mixtures were eluted and separated on a separation column (self-packed Picofrit C18 column, 75 μm ID x 40 cm, Tip ID 15 μm, New Objective, packed with Reprosil-Pur 120 C18-AQ, 3 μm, 120 Å, Dr. Maisch GmbH) using a linear gradient of 3% to 40% solvent B (in solvent A) over 120 min, 50% to 85% B (over 5 min) and 85% B (5 min) with the following solvent compositions: solvent A: water containing 0.1% (v/v) formic acid and solvent B: acetonitrile containing 0.1% (v/v) formic acid. The separation column was kept at 40 °C.

MS data were acquired in data-dependent MS/MS mode using a parallel accumulation-serial fragmentation (PASEF) method: The ion mobility scan range was set between 0.6 and 1.6 V s/cm² with a ramp time of 100 ms. 10 PASEF MS/MS scans were triggered per cycle (1.17 s) with a maximum of three precursors per mobilogram, the mobility-dependent collision energy was ramped linearly between 20 eV at an inverse reduced mobility ($1/k_0$) of 0.6 V s/cm² and 59 eV at 1.6 V s/cm². Target intensity per individual PASEF precursor was set to 20,000 with an intensity threshold of 2,500. Precursor ions in an $m/z$ range between 100 and 1700 with charge states ≥0+ and ≤5+ were selected for fragmentation. Active exclusion was enabled for 0.4 min (mass width 0.015 Th, $1/k_0$ width 0.015 V s/cm²).

## MS statistical analyses

For protein identification and feature detection, MS raw data were searched against the uniprot *Homo sapiens* database (version 06/23, 20,348 entries) using MaxQuant with default settings for Bruker TimsTOF data. This enables high peptide identification rates, individualized high mass accuracies and proteome-wide protein quantification[48]. Oxidation of methionine residues and acetylation of protein N-termini were set as variable modifications, and carbamido-methylation of cysteine residues was included as fixed modification. A maximum of two missed tryptic cleavage sites were considered for peptides. Peptides were quantified via label-free quantitation (LFQ). For relative protein quantification, LFQ intensities were processed using Perseus (version 2.0.7, www.maxquant.org). Proteins were filtered for group I with LFQ intensities for at least 2 out of 3 replicates for each sample or group II with LFQ intensities for all 3 replicates in at least one sample. After logarithmic transformation, missing LFQ intensities for protein group I were imputed by k-nearest neighbour (KNN) algorithm. For protein group II, missing values were imputed based on normal distributions around the lowest observed intensities. Statistical evaluation was done by Student's $t$ test and $\log_2$ fold changes between comparisons with an FDR < 0.05 were considered significant.

## Pathway analyses

Significant proteins identified from the statistical analyses of the MS experiments were evaluated regarding their subcellular location using the COMPARTMENT subcellular localization database[49]. Proteins located in the ER were selected and their interactomes were analyzed via the STRING database[50]. Pathway clusters were analysed by Gene Ontology (GO) enrichment analysis[51,52] and KEGG pathway enrichment analysis[53–55].

## Statistics and reproducibility

Intensities of protein bands in black-white Western blot images were quantified by ImageJ. The percentage cleaved was calculated as the fraction of the intensity for NTF bands over the total intensity for the monomeric receptor (cleaved + uncleaved). Percentage Endo H resistance was calculated as the percentage of the intensity for bands representing Endo H resistance over the sum of the intensities from Endo H-resistant and Endo H-sensitive bands of the monomeric receptors. Percentage cleavage contributed by membrane proximity was obtained as the difference of the percentage of cleavage between aGPCR-ECR-CalnTM$^{ER}$ and aGPCR-ECR-Fluo-CalnTM$^{ER}$. Signals from ELISA contributed by the expressed receptor were calculated by

subtracting the raw absorbance from the mean absorbance of cells transfected with the empty vector, and were represented as the percentage of the mean signals from WT. All statistical tests and the corresponding figures were generated in GraphPad Prism.

All datasets were first tested for normality by the Shapiro-Wilk test. The following list shows the statistical test chosen for post-normality testing (confidence interval: 95 %) and the number of replicates in each figure panel (N indicates biological replicates; n indicates technical replicates):

Figure 1c: Ordinary one-way ANOVA followed by a Tukey's post-hoc test. $N = 3$, $n = 4$.

Figure 1k: Unpaired two-tailed t-test. $N = 4$ for E2$^{WT}$-ECR-CalnTM$^{ER}$ and $N = 3$ for E2$^{WT}$-ECR-tdT-KDEL$^{ER}$.

Figure 2a: Kruskal-Wallis test followed by a Dunn's post-hoc test. $N = 3$.

Figure 2d: Ordinary one-way ANOVA followed by a Tukey's post-hoc test. $N = 3$.

Figure 2e: Unpaired two-tailed t-test. $N = 3$.

Figure 2f: Kruskal-Wallis test followed by a Dunn's post-hoc test. $N = 3$.

Figure 3c: Ordinary one-way ANOVA followed by a Tukey's post-hoc test. $N = 3$.

Figure 3d: Kruskal-Wallis test followed by a Dunn's post-hoc test. $N = 3$, $n = 4$.

Figures 3e and 3f: Two-way ANOVA, followed by Bonferroni's test with respect to different variants of the same timepoint. $N = 3$, $n = 3\text{-}4$.

Figure 3k: Kruskal-Wallis test followed by a Dunn's post-hoc test. $N = 6$, $n = 3\text{-}4$.

Figure 3l-n: Two-way ANOVA, followed by Bonferroni's test with respect to different variants of the same time point. $N = 3$, $n = 3\text{-}4$.

Figure 4c: Unpaired two-tailed t-test. $N = 3$ (E5, L3, D1, dCirl-E) or 4 (E2, G1, rL1).

Figure 4f: Kruskal-Wallis test followed by a Dunn's post-hoc test. $N = 3$.

Figures 5h and 5i: Ordinary one-way ANOVA followed by a Tukey's post-hoc test. $N = 3$.

All representative blots and microscopic images shown have been reproduced at least three times.

### Illustrations
All illustrations were created in Biorender and Adobe Illustrator.

### Reporting summary
Further information on research design is available in the Nature Portfolio Reporting Summary linked to this article.

## Data availability
The mass spectrometry proteomics data have been deposited to the ProteomeXchange Consortium via the PRIDE partner repository with the dataset identifier PXD056561. The uncropped scans for all Western blots shown in this study are available in Figshare repository [https://doi.org/10.6084/m9.figshare.27174246]. The processed data generated in this are provided in the Source data file. Source data are provided with this paper.

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

## Acknowledgements

This work was supported through grants of the Deutsche Forschungsgemeinschaft to T.L. and A.S. through CRC 1423 project number 421152132 (projects A06 and B03). A.S. acknowledges funding for mass spectrometers by the DFG (INST 271/404-1 FUGG and INST 271/405-1 FUGG), the Martin-Luther-University Halle-Wittenberg, and the region of Saxony-Anhalt (Center for Structural Mass Spectrometry). Y.K.C. and S.M. were funded by the Novo Nordisk Fonden (NNF23OC008432) during the manuscript revision. We thank Dirk Tänzler for excellent technical assistance.

## Author contributions

Y.K.C. designed, performed and analysed the experiments; prepared figures; wrote the manuscript with consent from all co-authors. Transgene cloning, protein expression, cell culture, RUSH and GCE assays. C.H.I. and A.S. performed LC-MS/MS and MS statistical analyses. L.Z. performed experiments involving microscopy. S.M. designed the experiments and analysed the data. T.L. conceived the study; designed the experiments; analysed the data; wrote the manuscript with consent from all co-authors.

## Funding

## Competing interests

The authors declare no competing interests.
