## [Transparent Peer Review file · Nature Communications]

Self-cleavage of the GAIN domain of adhesion G protein-coupled receptors requires multiple domain-extrinsic factors

Corresponding Author: Professor Tobias Langenhan

Version 0:

Reviewer comments:

Reviewer #1

(Remarks to the Author)

The responses from the authors to the reviewer's comments fully comply with the required rigor that would be expected from such a study. This reviewer finds the effort put in solving many raised issues as highly commendable. This manuscript needs no further improvement giving its scope.

Reviewer #2

(Remarks to the Author)

Reviewer #3

(Remarks to the Author)

This manuscript reports studies on the adhesion GPCR known as ADGRE2. The authors performed exquisitely detailed biochemical studies characterizing the processing of the self-cleaving GAIN domain of ADGRE2 as it moves through the Golgi and endoplasmic reticulum to the plasma membrane. The studies were rigorously performed, and the findings are novel and interesting. Moreover, in the revised version of this manuscript, the authors extended some of their studies on ADGRE2 to include six additional adhesion GPCRs in order to explore the generality of their findings. Overall, the authors were highly responsive to the reviewers' comments and have addressed all of the reviewers' concerns in a thorough manner. The textual changes and new data added by the authors have significantly strengthened the manuscript.
